# The Role of the Gluten-Free Diet in the Management of Seronegative Enteropathy

**DOI:** 10.3390/nu13114027

**Published:** 2021-11-11

**Authors:** Anna Szaflarska-Popławska

**Affiliations:** Department of Paediatric Endoscopy and Gastrointestinal Function Testing, Ludwik Rydygier Collegium Medicum in Bydgoszcz, Nicolaus Copernicus University in Torun, ul. Jagiellonska 13-15, 85-067 Bydgoszcz, Poland; aszaflarska@wp.pl; Tel.: +48-52-5854-888; Fax: +48-52-5854-799

**Keywords:** seronegative enteropathy, seronegative coeliac disease, gluten-free diet

## Abstract

The differential diagnosis and treatment of seronegative enteropathy, also termed seronegative villous atrophy (SNVA), is a clinical challenge. Although seronegative coeliac disease (CD) is a frequent cause of SNVA, the aetiology can include immune-mediated, inflammatory, infectious, and drug-related forms. As a misdiagnosis of SNVA can result in patients being unnecessarily placed on a lifelong strict gluten-free diet or even given incorrect immunosuppressive therapy, the aim of this paper is to provide an evidence-based and practical approach for the workup and management of SNVA.

## 1. The Definition of Seronegative Enteropathy

Seronegative enteropathy, also termed seronegative villous atrophy (SNVA), is characterised by the presence of a certain degree of small intestinal villous atrophy, but with the absence of coeliac serology [1]. Currently, the serological diagnosis of coeliac disease (CD) involves the testing of three types of antibodies: anti-tissue transglutaminase antibody (anti-TG2), anti-endomysial antibody (EmA), and antibody against deamidated gliadin peptides (anti-DGP) [2]. However, the precise criteria for a diagnosis of SNVA are vague. The 2021 American Gastroenterological Association Clinical Practice Update proposes the absence of all three antibody types [1], while some experts propose only anti-TG2 and EmA seronegativity [3,4], as noted in a few original works [5].

According to the current definition, certain cases do not meet the criteria for the diagnosis of SNVA: for example, the presence of isolated intraepithelial lymphocytosis of the small intestine mucosa (Marsh grade 1) in a seronegative patient [6] or selective IgA deficiency, i.e., where celiac-specific IgA antibodies are absent but specific IgG antibodies (IgG-seropositive coeliac disease) are present. The latter is also ruled out in patients receiving a gluten-free diet, those with a significant reduction in the supply of gluten, or those on immunosuppressive treatment [3]. Regular consumption of gluten, i.e., one to three slices of gluten bread per day for a period of ≥ six weeks, is needed to ensure reliable serological and histological test results for coeliac disease, i.e., to avoid false negative results [2].

## 2. Classification of Seronegative Enteropathy

Seronegative coeliac disease (SNCD) should be distinguished from non-coeliac seronegative villous atrophy (non-CD SNVA) based on aetiology.

## 3. Seronegative Coeliac Disease (SNCD)

SNCD is a form of celiac disease (CD) that occurs in patients with a confirmed genetic predisposition. Assuming that other causes of enteropathy have been excluded, it is characterised by various degrees of villous atrophy of the small intestine mucosa, and the absence of serum IgA and IgG anti-TG2 and EmA. In addition, patients demonstrate clinical and histological improvement after the introduction of a gluten-free diet. According to the 2021 American Gastroenterological Association Clinical Practice Update, the diagnosis of SNCD requires negative results for both the serological test and with regard to IgA and IgG class anti-DGP [1]. The degree of villous atrophy of the small intestinal mucosa should be assessed according to the Marsh–Oberhuber classification shown in Table 1 [7]. 

However, following the elimination or limitation of dietary gluten in CD patients, or the use of glucocorticosteroids or immunosuppressive drugs, such tests may not identify any serological markers of CD [3]. These patients may also lack specific antibodies in early stage CD, as patients with less severe atrophic changes in the mucosa of the small intestine are significantly more likely to demonstrate a lack of EmA than patients with flat mucosa [8]. In addition, serum anti-TG2 antibodies are often absent in first-degree relatives of CD patients [9] and in up to 50% of Caucasian patients with dermatitis herpetiformis [10]. Seronegativity is also common in patients with CD resistant to dietary treatment (i.e., refractory coeliac disease) [11]. 

Although SNCD is rare, accounting for 1.7–5% of all cases of coeliac disease, it has been estimated that one-third of all cases of SNVA in adults can be classified as seronegative [11]. In children, SNCD is extremely rare [12] and Volta et al. [13].

Certain genetic and clinical features can be used to differentiate patients with seropositive and seronegative coeliac disease; however, as such studies have been performed on small groups and many have used inconsistent classifications for seronegativity, their findings should be regarded with caution. Volta et al. [13] report that SNCD was diagnosed significantly later than the seropositive form, that the course was more commonly associated with classic symptoms (diarrhoea, malabsorption, weight loss) and advanced histopathological changes in the small intestine, and that it was frequently associated with other autoimmune diseases. Dore et al. [14] report that patients with SNCD were more likely than seropositives to have mild histopathological changes classified as grade 1 according to Marsh. However, it should be remembered that according to the current definition, isolated intraepithelial lymphocytosis should not be classified as enteropathy. In the largest study to date involving 227 adult patients previously diagnosed with SNVA, Schiepatti et al. found that SNCD had more severe symptoms at diagnosis, nearly 11 times higher risk of complications, and 2 times higher risk of mortality than seropositive CD [15]. 

## 4. Non-Coeliac Seronegative Villous Atrophy (Non-CD SNVA)

The differential diagnosis for non-CD SNVA is broad and includes autoimmune and immune-mediated, infectious and probably infectious, inflammatory, neoplastic, and iatrogenic causes (Table 2).

According to the few studies available on the subject, the frequency of occurrence of SNVA varies considerably, particularly between adult and child populations. CD is diagnosed in 28–45% of adults with SNVA [5,13,40]; among the remainder, the most common causes of non-CD SNVA are believed to be *inter alia* medication-related enteropathy and infectious causes, including giardiasis and small intestinal bacterial overgrowth, peptic duodenitis, and collagenous sprue [5,13,40,41]. In the paediatric population, the most common causes of SNVA are believed to be inflammatory bowel disease and infections, such as *Giardia lamblia* and *Helicobacter pylori* [12,17]. In up to 1/3 of patients, the cause of SNVA cannot be determined [5,12,13,17,40,41]. The main causes of SNVA in children and adults are summarised in Table 3.

## 5. Histopathological Evaluation of Seronegative Enteropathy 

The histopathological criterion of SNVA is the presence of atrophy of the villi of the small intestinal mucosa. An accurate diagnosis requires the evaluation of four to six biopsies, collected from the appropriate sites of the small intestine (bulb and extra-bulbar part of the duodenum); these should be correctly oriented on the filter paper, properly fixed, and stained. Assessment should be performed by an experienced histopathologist using a commonly accepted microscopic classification. The expert opinion of the American Gastrointestinal Association [1], the sole existing opinion, is that the histopathological assessment of SNVA should be based on the Corazza–Villanacci classification.

The presence of atrophic changes in the mucosa of the small intestine limited only to the duodenal bulb should be interpreted with great caution. In 10% of cases, microscopic changes are restricted to the first part of the duodenum, a condition known as ultra-short coeliac disease [42]. In addition, deformation of the intestinal villi in the mucosal areas of overlying Brunner glands, lymphoid follicles, and within digestive lesions may be misjudged by a less experienced histopathologist as enteropathy [43]; a diagnosis of enteropathy without intraepithelial lymphocytosis makes the diagnosis of celiac disease unlikely [44]. As different centres may reach different conclusions following the histopathological evaluation of mucosa biopsies [45], all uncertain diagnoses should be verified in an expert centre. 

Furthermore, in addition to histological assessment of any atrophic changes identified in small intestine biopsies, it is important to confirm the presence of features specific to enteropathy of non-coeliac aetiology. In both seropositive and seronegative coeliac disease, intestinal villous atrophy is accompanied by crypt hyperplasia and intraepithelial lymphocytosis [44]. Histopathological characteristics suggestive of enteropathy associated with Crohn’s disease include the presence of a lymphoplasmacytic infiltrate with neutrophilic cryptitis adjacent to histologically normal crypts, as well as usually poorly formed, non-necrotizing granulomas; however, the latter are infrequently seen in both adults and children [46]. 

In addition, autoimmune enteropathy with small intestinal villous blunting and crypt hyperplasia histologically mimics CD. However, autoimmune enteropathy is distinguished by deep crypt lymphocytosis with minimal surface lymphocytosis and apoptosis, and a lack of goblet and Paneth cells [47]. 

Common variable immune deficiency (CVID)-related enteropathy shares some histological features with CD: small intestinal villous atrophy, crypt hyperplasia, and intraepithelial lymphocytosis are present in both conditions. However, the absence of plasma cells or their smaller number and graft-versus-host-disease-like apoptosis lesions are strongly suggestive of CVID-related enteropathy rather than CD [48]. Furthermore, while villous atrophy may be central to the histopathology of both CD and collagenous sprue, the latter is characterized by the presence of distinct subepithelial collagen bands (usually >20 μm). It is worth mentioning that subepithelial collagen deposition may be observed in rare cases, and this may be a consequence of another disease, such as medication-related enteropathy or CD [49]. In addition to the rarely observed atrophy of the intestinal villi, Whipple’s disease is also characterised by PAS-positive macrophagic infiltration of the lamina propria, which also represents the diagnostic cornerstone of the disease [50]. 

Non-coeliac enteropathy is also indicated by villous blunting accompanied by prominent mucosal eosinophilia, typically seen in food allergy and eosinophilic gastroenteritis. Histologic criteria usually suggested for the diagnosis of eosinophilic gastroenteritis include an increased eosinophil count higher than 20 eos/hpf in the duodenum and 56 eos/hpf in the ileum [36]. In addition, the presence of dense mucosal eosinophilia, often with sheets of eosinophils and usually not accompanied by epithelial lymphocytosis, especially in children may indicate food allergy or *Giardia lamblia* infection [20,51]. On rare occasions, some grade of villous atrophy with prominent neutrophilic inflammation, but with a normal lymphocyte count, may be seen in peptic duodenitis in both *Helicobacter pylori*-positive and -negative patients; this is often accompanied by gastric metaplasia and Brunner’s gland hyperplasia [17].

## 6. HLA Testing

Seronegative celiac disease is the most common cause of SNVA in adults. Both seropositive and seronegative CD only occur in genetically predisposed individuals; hence, a lack of genetic predisposition to the development of celiac disease practically excludes a positive diagnosis. A patient with SNVA should therefore be tested for HLA DQ2.5 (DQA1*0501, DQB1*0201), HLA DQ2.2 (DQA1*0201, DQB1*0202), HLA DQ8 (DQA1*03, DQB1*0302), and HLA DQ7.5 (DQA1*05, DQB1*0301). A lack of DQ heterodimers or of one-half of them is sufficient to exclude a diagnosis of SNCD [52]; in such cases, there is a need to identify the causes of damage to the small intestine not associated with CD. 

As the HLA DQ polymorphism significantly influences the likelihood of CD diagnosis, genetic testing can be used to not only determine a predisposition to CD but also to estimate the risk of disease. In patients with low genetic predisposition, causes of intestinal villus atrophy other than CD should be taken into account [53].

## 7. Medical History in Seronegative Enteropathy

In patients with SNVA, a detailed medical history allows more focussed diagnostic procedures to be used and limits the need for expensive and often poorly available laboratory testing. A number of drugs can cause drug-induced enteropathy, such as non-steroidal anti-inflammatory drugs (NSAIDs) and angiotensin II receptor blockers, as well as azathioprine, methotrexate, and mycophenolate mofetil [27,28,29,30,31]. As some forms of SNVA unrelated to gluten can develop as a consequence of an infection, a history of recent travel and correlation with other infectious symptoms should be carefully evaluated. For example, a recent visit, particularly one longer than a month, to a region between 30° north and 30° south of the equator, especially Cuba, Haiti, Puerto Rico, the Dominican Republic, India, Vietnam, Burma, Malaysia, and Indonesia, may indicate tropical sprue [46] while a recent history of acute self-limiting diarrhoea may suggest post-viral enteropathy [38]. Drinking unboiled water from streams, rivers, and lakes, or staying in developing countries characterised by poor sanitary and epidemiological conditions are known risk factors for *Giardia lamblia* infection [38]; in addition, abdominal surgery, diseases associated with intestinal motility disorders, and the use of drugs that inhibit gastric secretions are risk factors for SIBO [54]. A history of recurrent opportunistic infections often associated with weight loss and night sweats in patients with SNVA should prompt a workup for HIV-enteropathy [48]. Alternatively, intermittent watery diarrhoea caused by the Gram-positive bacterium *Tropheryma whipplei*, in a patient with long-standing seronegative arthritis, weight loss, fever, and lymphadenopathy is characteristic of late-course Whipple’s disease [50]. 

Common variable immune deficiency (CVID) should be suspected in patients with SNVA and recurrent infections, mostly involving the respiratory and gastrointestinal tract and/or coexisting autoimmune diseases, particularly haemolytic anaemia and immune thrombocytopenia purpura. However, the differentiation between CVID-related enteropathy and CD in a patient with CVID may be challenging. It is worth mentioning that HLA-DQ2/DQ8 negativity and non-responsiveness to a gluten-free diet might also play valuable roles in ruling out CD in such patients [48]. 

## 8. Blood and Stool Tests in SNVA

Certain serological tests may be useful in determining the cause of SNVA. The presence of anti-enterocyte or anti-goblet cell antibodies would support a diagnosis of autoimmune enteropathy. However, it should be kept in mind that these antibodies are not pathognomonic for autoimmune enteropathy, as they may be detected in other inflammatory diseases. Moreover, as they are tested by immunofluorescence, the results are observer dependent [16]. Anti-*Saccharomyces cerevisiae* antibody (ASCA) and perinuclear anti-neutrophil cytoplasmic antibody (p-ANCA) tests may be helpful for the diagnosis of inflammatory bowel disease as a cause of SNVA [55]. A marked decrease in total serum IgG and a decrease of at least one of the isotypes of IgG or IgA are characteristic of common variable immune deficiency [48]. 

HIV enteropathy and enteropathy in the course of *Mycobacterium tuberculosis* infection may be diagnosed based on the presence of HIV antibodies and QuantiFERON-TB, respectively. In addition, both small intestinal bacterial overgrowth and *Helicobacter pylori* infection can be confirmed by breath testing: a valuable and non-invasive diagnostic method [4]. Furthermore, stool tests should be carried out for *Giardia antigen*, ovum and cyst microscopy for parasitic-related enteropathy, and calprotectin for inflammatory bowel disease [4]. A proposed diagnostic procedure for patients with SNVA is presented in Figure 1.

## 9. Treatment of Seronegative Enteropathy

There is no doubt that in the case of non-CD SNVA patients with identified aetiology, treatment and monitoring procedures should be selected with the aim of managing the cause of the enteropathy. Appropriate causal treatment often resolves the clinical and histopathological symptoms of enteropathy, and a biopsy is not always necessary for confirming microscopic remission [1]. 

For patients with SNVA who demonstrate a genetic predisposition for CD, and in whom other causes of enteropathy have been excluded, a gluten-free diet is recommended. If clinical and histological remission is observed after excluding gluten from the diet, this would confirm a diagnosis of seronegative CD; this would require the patient to strictly follow a gluten-free diet for the rest of their life. Excluding gluten from the diet in patients with CD, either seropositive or seronegative ones, often results in the resolution of clinical symptoms and histological remission [14]. However, if it is not possible to confirm compliance with a gluten-free diet in a patient with SNCD using serological tests, a biopsy of the small intestine should be performed, as clinical improvement alone is not sufficient evidence of treatment efficacy. It is arbitrarily assumed that such biopsies of the small intestine should be performed after about 12 months of following a gluten-free diet, or earlier in patients with a severe clinical course [1]. Some adult patients with CD, either seropositive or seronegative, do not achieve clinical and/or histological remission after 12 months of a gluten-free diet; such cases meet the criteria of refractory coeliac disease and should be treated accordingly [56]. 

In the absence of an identified cause of SNVA in a patient with no genetic predisposition to CD, which may affect up to one-third of patients, the decision regarding the treatment method should depend on the clinical condition of the patient. In those with a good clinical condition, a wait-and-see strategy should be followed. In this case, systematic monitoring of the patient, including histological testing, may be considered, as some patients experience spontaneous healing of the mucosa of the small intestine. Aziz et al. [5] reported spontaneous resolution of intestinal villous atrophy in 72% of untreated idiopathic SNVA patients within nine months of diagnosis. In patients with SNVA of undetermined aetiology and whose clinical condition does not allow a wait-and-see approach, or in patients in whom the inclusion of a gluten-free diet was unsuccessful, including those with refractory coeliac disease, the first treatment option should be corticosteroid treatment, either local (open-capsule budesonide) or systemic (prednisone). Corticosteroids should be generally continued for several weeks to months and then slowly tapered. To treat patients who are steroid refractory, several second-line therapies are proposed, including immunosuppressive agents (azathioprine) [57]. The rational approach to SNVA of unknown aetiology is presented in Figure 2.

SNVA is a histologic finding that can occur in a wide range of aetiologies. It can be misdiagnosed due to the misinterpretation of small intestinal histopathology, insufficient serologic testing, IgA deficiency, concomitant immunosuppressive medications or initiation of a gluten-free diet. In adults SNCD is the most common cause of SNVA. The diagnosis of SNCD requires compatible HLA genetics and both clinical and histological improvement on a gluten-free diet. A gluten-free diet is the main treatment for SNCD patients and in stable patients in whom no alternative aetiology has been found and coeliac disease cannot be ruled out. 

## Figures and Tables

**Figure 1 nutrients-13-04027-f001:**
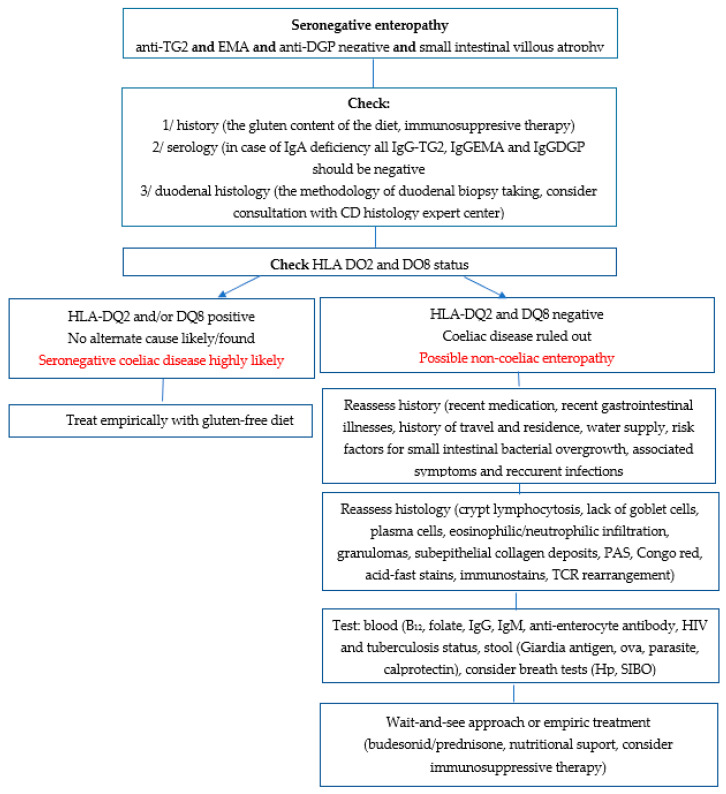
A proposed rational approach to diagnosing patients with seronegative enteropathy.

**Figure 2 nutrients-13-04027-f002:**
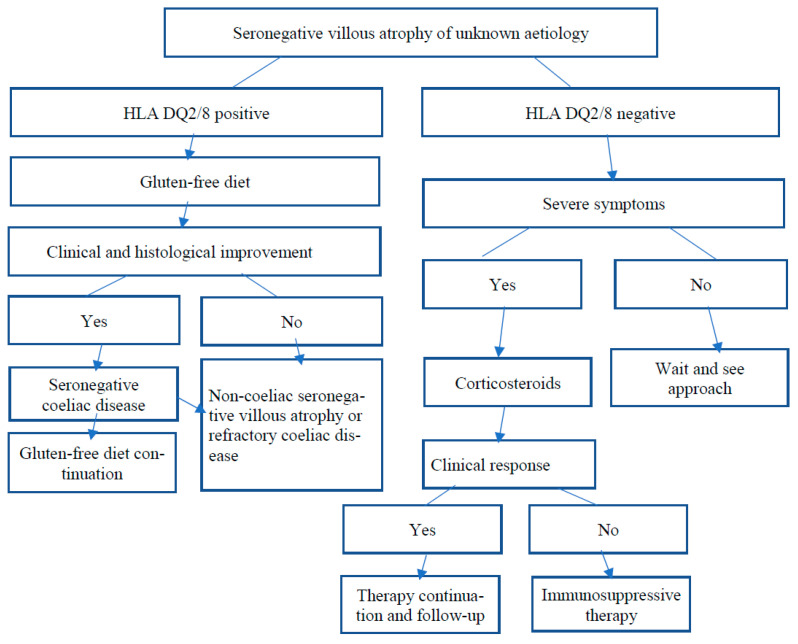
A stepwise algorithm for the treatment of seronegative villous atrophy of unknown aetiology.

**Table 1 nutrients-13-04027-t001:** The modified Marsh–Oberhuber classification [7].

	Number of Intraepithelial Lymphocytes per 100 Enterocytes	Crypt Hyperplasia	Villous Atrophy
Marsh 0	<30/100	−	-
Marsh 1	>30/100	−	-
Marsh 2	>30/100	+	-
Marsh 3a	>30/100	+	mild (partial)
Marsh 3b	>30/100	+	moderate (subtotal)
Marsh 3c	>30/100	+	total
Marsh 4	<30/100	−	total

**Table 2 nutrients-13-04027-t002:** The most common causes of non-coeliac seronegative enteropathy (non-CD SVNA).

Aetiology of Non-Coeliac Seronegative Villous Atrophy
Autoimmune and immune-mediated diseases
Autoimmune enteropathy [16]
Crohn’s disease [17]
Collagenous sprue [18]
Common variable immune deficiency (CVID) [19]
Infectious or probably infectious diseases
*Giardia lamblia* infection [20]
*Helicobacter pylori* infection [17]
Post-viral enteropathy [21]
Tuberculosis [22]
Small intestinal bacterial overgrowth [23]
HIV-enteropathy [24]
Whipple disease [25]
Tropical sprue [26]
Iatrogenic causes
Medication (angiotensin II receptor blockers [27], mefenamic acid [28], azathioprine [29], methotrexate [30], mycophenolate mofetil [31])
Chemotherapy [32]
Radiotherapy [33]
Graft versus host disease [34]
Inflammatory disease
Peptic duodenitis [35]
Eosinophilic gastroenteritis [36]
Neoplastic
Small intestinal lymphoma [37]
Other
Amyloidosis [38]
Malnutrition [39]
Food allergy (cow’s milk, soy) [20]

**Table 3 nutrients-13-04027-t003:** Frequencies of causes of seronegative enteropathy.

First Author and Publication Date	Study Population	Causes of Enteropathy (%)	Limitations
Pallav K, 2012 [40]	30 non-coeliac adults, 20–76 years (median age 54.5 years)	Peptic duodenitis (16.6%) Collagenous sprue (10%) Eosinophilic gastroenteritis (6.7%) Bacterial overgrowth (10%) Viral gastroenteritis (6.7%) Crohn’s disease (6.7%) Tropical sprue (3.3%) Food allergy (3.3%) Common variable immunodeficiency (3.3%) Unclassified sprue (33.3%)	Seronegative coeliac disease excluded, medication-related enteropathy not represented
DeGaetani, M, 2013 [39]	72 adults, 29–85 years (mean 59 years)	Seronegative coeliac disease (28%) Medication-related enteropathy (26%) Unclassified sprue (14%) Common variable immunodeficiency (6%) Autoimmune enteropathy (4%) Giardiasis (4%) CD4+ T cell lymphoma (4%) Tropical sprue (4%) Bacterial overgrowth (3%) Collagenous sprue (3%) Enteropathy-associated T cell lymphoma (1%) Crohn’s disease (1%) Extensive gastric metaplasia (1%)	Different patterns of testing for the workup of seronegative enteropathy cases, poor HIV testing
Volta U, 2016 [13]	31 adults	Seronegative coeliac disease (45.2%) Giardiasis (19.4%) Common variable immunodeficiency (16.1%) Autoimmune enteropathy (9.7%) Small intestinal bacterial overgrowth (3.2%) Olmesartan enteropathy (3.2%) Eosinophilic enteritis (3.2%)	Serology based only on anti-TG2 and EMA testing in IgA-non-deficient patients, and on anti-TG2 in IgA-deficient patients
Aziz I, 2017 [5]	200 adults, mean age 51.2 ± 17.6 years	Seronegative coeliac disease (31%) Infectious enteropathy (27%) Idiopathic enteropathy (18%) Peptic duodenitis (11.5%) Drug-related enteropathy (6%) Crohn’s disease (3%) Systemic immune-mediated (2%) Radiation enteritis (0.5%) Eosinophilic enteritis (0.5%)	No anti-DGP testing, IgG-seropositive coeliac disease patients included
Gustafsson I, 2020 [12]	40 children from two cohorts (Finnish and Romanian)	Giardiasis (25%) *Helicobacter pylori* infection (15%) Inflammatory bowel disease (17.5%) Cow’s milk allergy (15%) Malnutrition (7.5%) Autoimmune enteropathy (2.5%) Rotavirus infection (2.5%) No diagnosis (15%)	Non-systemic use of serology in the older series
Mandile R, 2021 [16]	64 children, mean age 5.9 years	Inflammatory bowel disease (32.8%) Gastro-oesophageal reflux disease (18.8%) Food allergy (12.5%) Infectious enteropathy (10.9%) Immunodeficiency (4.7%) Short bowel syndrome (4.7%) Congenital diarrhoea (3.1%) Other/inconclusive diagnosis (12.5%)	Samples reviewed by different pathologist; immunohistochemistry staining was not performed for all patients
Schiepatti A, 2021 [15]	227 adults previously diagnosed with seronegative villous atrophy	Seronegative coeliac disease (37%) IgG-seropositive coeliac disease (21.1%) Coeliac disease not confirmed/excluded (lack of a follow up biopsy) (17.6%) Poor orientation or histological misinterpretation of biopsies (17.6%) Gluten-free diet or immunosupressive therapy at time of diagnosis (7.9%) Idiopathic enteropathy (4%) Collagenous sprue (0.9%) Crohn’s disease (0.4%) Autoimmune enteropathy (4%) Tropical sprue (0.4%)	IgG-seropositive coeliac disease included

## Data Availability

Not applicable.

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
