# Peer review of "The Role of the Gluten-Free Diet in the Management of Seronegative Enteropathy"

_nutrients, 2021, doi:10.3390/nu13114027_

Round 1
Reviewer 1 Report
Dear Author,
first of all, I wish to thank you for giving me the opportunity to read this your manuscript, submitted for publication in Nutrients.
I have few comments and suggestions. I hope they will be useful.
- line 31: Marsh grade 1. Please, consider a table with Marsh's classification. Many readers may not know it.
- Table 2 should be expanded and updated.
- Figure 1 : its contents are misread.
- References : they could be updated. Please, consider this suggestion.
Author Response
Dear Sir/Madam
Firstly, I would like to thank you for giving me the opportunity to submit a revised draft of my manuscript entitled "The role of gluten-free diet in the management of seronegative enteropathy" . I appreciate the time and effort that you have dedicated to providing your valuable feedback on my manuscript. I have been able to incorporate changes to reflect all suggestions. Here is a point-by-point response to your comments and suggestions.
Comment 1: Line 31 - Marsh grade 1 - Please consider a table with Marsh's classification. Many readers may not know it.
Response: The modified Marsh-Oberhuber classification (tab. 1) has been incorporated into the text.
Comment 2: Table 2 should be expanded and updated.
Response: The results of the newly published study aiming to evaluate the clinical spectrum and natural history of patients previously diagnosed with seronegative villous atrophy by Schiepatti et al. (Aliment Pharmacol Ther 2021, 54, 1278-89) were incorporated into the table 3 (previously table 2).
Comment 3: Figure 1 - its contents are misread.
Response: The fond size in the figure 1 has been increased to make it easier to read.
Comment 4: References: they could be updated
Response: Two older publications (previously numbered 14 and 25) have been replaced by papers published more recently (2019 and 2021). One publication (2021) has been added.
The paper has been corrected by a native speaker, who is experienced with medical language and moderate changes have been made.
II look forward to hearing from you and to respond to any further questions and comments you may have.
Sincerely, Anna Szaflarska-Popławska

Reviewer 2 Report
This is a great manuscript very helpful in diagnosis of antibody negative celiac disease vs. other enteropathies. I have no recommendations.
The main question of the manuscript is diagnosis and classification of antibody negative celiac disease versus other etiologies of enteritis. This is a very confusing topic that the authors seek to clarify which they do well. It is a very timely paper in that recently antibody testing has improved giving the diagnostician more options for testing and diagnosis. I feel the paper is novel there is a diagnostic rubric for sorting the different etiologies of enteritis out. The paper is clear and easy to read despite the confusing topic. Overall, I really enjoyed reading and reviewing the paper and I feel it contributes nicely to the body of literature available.
Author Response
Dear Sir/Madam,
Thank you for giving me the opportunity to submit a revised draft of my manuscript entitled "The role of the gluten-free diet in the management of seronegative enteropathy". I appreciate the time and effort that you have dedicated to providing your valuable feedback on my manuscript. As suggested my manuscript has been corrected by a native speaker who is also experienced with medical language and some changes have been made.
I look forward to hearing from you and to respond to any questions you may have.
Sincerely, Anna Szaflarska-Popławska
